# Regulation of Epithelial–Mesenchymal Plasticity by the E3 Ubiquitin-Ligases in Cancer

**DOI:** 10.3390/cancers12113093

**Published:** 2020-10-23

**Authors:** Andrea Rodríguez-Alonso, Alba Casas-Pais, Daniel Roca-Lema, Begoña Graña, Gabriela Romay, Angélica Figueroa

**Affiliations:** 1Epithelial Plasticity and Metastasis Group, Instituto de Investigación Biomédica de A Coruña (INIBIC), Complexo Hospitalario Universitario de A Coruña (CHUAC), Sergas, Universidade da Coruña (UDC), 15006 A Coruña, Spain; andrea.rodriguez.alonso@sergas.es (A.R.-A.); alba.casas.pais@sergas.es (A.C.-P.); Daniel.roca.lema@sergas.es (D.R.-L.); Gabriela.Romay.Cousido@sergas.es (G.R.); 2Clinical Oncology Group, Instituto de Investigación Biomédica de A Coruña (INIBIC), Complexo Hospitalario Universitario de A Coruña (CHUAC), Sergas, Universidade da Coruña (UDC), 15006 A Coruña, Spain; Begona.Grana.Suarez@sergas.es

**Keywords:** epithelial–mesenchymal plasticity, posttranslational regulation, ubiquitination, E3 ubiquitin-ligases, targeted protein degradation, drug targets, cancer

## Abstract

**Simple Summary:**

Cancer progression eventually may lead to metastasis, which is responsible for at least 90% cancer-related deaths. The epithelial-to-mesenchymal transition plays a critical role in promoting metastasis and is considered a new target in anticancer drug discovery. The E3 ubiquitin-ligases responsible for targeted protein degradation emerge as molecularly defined targets involved in epithelial-to-mesenchymal transition. This review highlights the novel E3 ubiquitin-ligases responsible for targeted protein degradation involved in epithelial-to-mesenchymal transition, a field that promises to be one of the greatest innovation in small-molecule drug discovery.

**Abstract:**

The epithelial–mesenchymal plasticity (EMP) is a process by which epithelial cells acquire the ability to dynamically switch between epithelial and mesenchymal phenotypic cellular states. Epithelial cell plasticity in the context of an epithelial-to-mesenchymal transition (EMT) confers increased cell motility, invasiveness and the ability to disseminate to distant sites and form metastasis. The modulation of molecularly defined targets involved in this process has become an attractive therapeutic strategy against cancer. Protein degradation carried out by ubiquitination has gained attention as it can selectively degrade proteins of interest. In the ubiquitination reaction, the E3 ubiquitin-ligases are responsible for the specific binding of ubiquitin to a small subset of target proteins, and are considered promising anticancer drug targets. In this review, we summarize the role of the E3 ubiquitin-ligases that control targeted protein degradation in cancer-EMT, and we highlight the potential use of the E3 ubiquitin-ligases as drug targets for the development of small-molecule drugs against cancer.

## 1. Introduction

Epithelial–mesenchymal plasticity (EMP) is an important reversible program characterized by the ability of epithelial cells to dynamically switch between different phenotypic cellular states. This process encompasses the epithelial–mesenchymal transition and the mesenchymal–epithelial transition, bearing important implication for tumor progression and metastasis, therapeutic resistance and cancer stem cell properties [1]. Tumor cells that undergo epithelial–mesenchymal transition lose cell–cell and cell–extracellular matrix interactions, leading to cell migration and invasion. Although the repercussion of the transcriptional and post-transcriptional regulation of the EMP has been well documented, post-translational regulation has recently emerged as a critical regulator of this program. In this review, we focus on the regulation of the cancer-EMP by targeted protein degradation induced through ubiquitination, a post-translational modification that regulates protein turnover. Specifically, we highlight the crucial role of the E3 ubiquitin-ligases in cancer-EMP, enzymes responsible for the recognition of specific target proteins targeted for degradation. Moreover, we highlight the potential use of the E3 ubiquitin-ligases as promising drug targets for the development of small molecule drugs against cancer. 

## 2. Epithelial–Mesenchymal Plasticity and Tumor Progression 

Most cancer-related deaths are due to metastasis, a complex process in which cancer cells are able to disseminate from a primary tumor to grow at distant secondary sites. In epithelial-derived carcinomas, cancer cells acquire the ability to invade and metastasize as a result of epithelial-to-mesenchymal transition (EMT), a cellular program that was firstly described in embryonic development [2,3,4]. During EMT, epithelial cells undergo phenotypic transformation into mesenchymal cells, due to the loss of apical-basal polarity, loss of cell junctions, and the reorganization of the actin cytoskeleton, which in turn causes the acquisition of migratory and invasive capabilities [1]. EMT has also been linked to apoptosis suppression and resistance to chemotherapy and immunotherapy, as well as the acquisition of stem cell characteristics, conferring selective advantage to cancer cells [5,6,7,8,9,10]. EMT is not an irreversible process and the resulting mesenchymal cells can revert to epithelial phenotype by mesenchymal-to-epithelial transition (MET) [4]. Indeed, MET usually happens when cancer cells reach secondary sites. They revert to an epithelial phenotype to enhance cell adhesion to the substrate in the distant organ and form metastasis. In a tumor, cancer cells usually execute an incomplete EMT program. Therefore, EMT is not a binary process but there are different intermediate states [11,12] that are needed for the metastatic colonization [13,14]. This phenotypic plasticity allows cancer cells to metastasize and seed new tumors at distant sites [15]. Indeed, there is a variety of intermediate cell phenotypes between epithelial and completely mesenchymal states. This is known as ‘epithelial–mesenchymal plasticity’ (EMP) and describes the ability of cells to adopt mixed epithelial/mesenchymal (E/M) features and to interconvert between both phenotypic states (Figure 1).

In normal epithelium, adhesions between epithelial cells are normally mediated by cell–cell contacts including adherens junctions (AJs), gap junctions, tight junctions (TJs) and desmosomes [16,17]. During the EMT process, cells lose apical-basal polarity and cell–cell junctions disappear, which in turn causes the acquisition of front-back cell polarity and a mesenchymal phenotype, as well as motile capacity. Moreover, the induction of the EMT program also leads to the reorganization of the actin cytoskeleton, as well as the degradation of the extracellular matrix (ECM) and the basement membrane. Degradations of the physical barriers result in the acquisition of invasive capacity, further allowing dissemination to distant tissues [1,18,19]. Perhaps the best characterized molecular hallmark of EMT is the loss of E-cadherin at cell–cell contacts. E-cadherin is the prototype and best characterized member of the classical family of cadherins, which are the main components of the transmembrane domain of adherens junctions [3,17]. E-cadherin is reported as a tumor suppressor, and its loss has been described as a crucial alteration in the development of epithelial cancers. E-cadherin downregulation not only causes the disruption of cell–cell adhesions, but it is also associated with increased invasion and metastasis. Therefore, the loss of the epithelial marker E-cadherin is a characteristic feature of EMT and also a hallmark of tumor malignancy [20,21]. However, EMT entails a profound cellular reorganization as a result of the downregulation of another epithelial cell marker, such as cytokeratins, as well as the upregulation of mesenchymal ones, including N-cadherin and vimentin. Changes in integrin expression also occur during the EMT process, and these are responsible for the remodeling of the interactions between cells and the ECM. Thus, during the acquisition of mesenchymal features, cells downregulate some epithelial integrins, such as α6β4, and increase the expression of others, such as α1β1 or α2β1 [18,22]. Given the well-recognized role of EMT in tumor progression, metastasis, stem-cell characteristics and therapy resistance, it has emerged as a promising therapeutic target for anticancer therapy [23].

## 3. Molecular Mechanisms of EMT in Cancer: Role of Targeted Protein Degradation by Ubiquitination

The suppression of the epithelial phenotype as well as the subsequent activation of the mesenchymal phenotype that takes place during the EMT program is tightly controlled at different levels: transcriptionally, post-transcriptionally and post-translationally, all with important clinical implications [24,25]. EMT is transcriptionally regulated by EMT-inducing transcription factors (EMT-TFs). These EMT-TFs include Snail (also known as Snai1), Slug (also called Snai2), Twist1, Zeb1 and Zeb2, which mediate the transcriptional repression of E-cadherin [18,26] and are considered master regulators of the expression of epithelial and mesenchymal genes. In addition to the effect of these TFs on E-cadherin repression at the transcriptional level, they are also implicated in the simultaneous repression of other junctional proteins, such as claudins, cytokeratins or desmosomes, facilitating the general dedifferentiation program [26,27]. The transforming growth factor-β (TGFβ) family is the main inducer of EMT during cancer progression. This action is in part mediated through the SMAD activation and the subsequent control of EMT-TFs. TFG-β family proteins can also lead to EMT activation through Smad-independent pathways [18]. The TGF-β Smad-dependent pathway leads to the induction of EMT by the activation of Smad2 and Smad3. Once activated, they form a complex in the nucleus with Smad4, thus regulating the expression of epithelial and mesenchymal genes. Conversely, EMT activation during the Smad-independent signaling response takes place through the activation of Erk MAP kinases, Rho GTPases or the PI3 kinase/Akt pathway [16,28]. 

On the other hand, posttranscriptional regulation has also been reported to regulate EMT. EMT-related genes are regulated at a posttranscriptional level by small non-coding RNAs or microRNAs and by RNA-binding proteins (RBPs) [29,30,31]. Remarkably, several RBPs involved in different post-transcriptional steps during EMT include: AKP8, ESRP1, Zeppo1, SF2/ASF, PSF (splicing), Sam68 (non-mediated decay), hnRNPE1 (translation) and TTP, HuR (mRNA turnover) [29,32]. Moreover, several micro-RNAs have emerged as key players during EMT in cancer. Micro-RNAs are small (~22-nt), single-stranded, non-coding RNAs that specifically bind mRNAs targets, thus promoting their degradation or inhibiting their translation [33]. Several miRNAs have been reported to play a role in EMT. Perhaps the miR-200 family, consisting of five members (miR-200a, -200b, -200c, -141, -429), and miR-205 are the best-documented miRNAs that regulate EMT by targeting ZEB1 mRNA [34,35,36]. Additionally, miR-29b, miR-30a and miR-203 repress SNAIL1 reverting the EMT program [18,37,38]. 

In the last decades, the control of the EMT at transcriptional, post-transcriptional and posttranslational levels (by phosphorylation, glycosylation or proteolysis) [39,40] has been very well documented [18,29], therefore, in this review, we will not look at this in depth. In the present review, we will go deeper into the impact of the targeted protein degradation by ubiquitination on EMT.

### 3.1. Ubiquitination Process

Ubiquitination is a reversible post-translational modification that targets specific proteins to different pathways and fates. It consists of the conjugation of a 76-amino acid ubiquitin molecule to a target protein controlling its abundance, localization, interactions or activity. Based on how the protein substrate is tagged by ubiquitin, this process can be classified as: monoubiquitination, when there is an addition of a single ubiquitin moiety to a target protein; multi-monoubiquitination when several ubiquitin molecules bind in different lysine residues of the same protein; and polyubiquitination, when subsequent rounds of ubiquitination in the same lysine lead to the formation of polyubiquitin chains. Ubiquitination controls target protein degradation by the proteasome, the lysosome or autophagy, but it can also control transcription, DNA repair or localization [41]. Recently, the impact of the targeted protein degradation mediated by polyubiquitination in cancer has been well-documented. For this ubiquitination reaction, sequential action of three different enzymes is required: the ubiquitin-activating enzyme (E1), responsible for ubiquitin activation in an ATP-dependent manner, the ubiquitin-conjugating enzyme (E2), which transfers the activated ubiquitin from the E1 enzyme to the E3 ubiquitin-ligases (Figure 2). The E3 ubiquitin-ligase enzyme is responsible for specific binding of ubiquitin to the target substrate [42,43]; the mechanism used to transfer ubiquitin to the target depends on the specific domain they contain. In that sense, substrate specificity of the ubiquitin proteasome system relies on the E3 ubiquitin ligases, which are larger in number in comparison with E1 and E2 enzymes. E3 ubiquitin-ligases receive the activated ubiquitin from the E2 conjugating enzyme and subsequently transfer it to a specific substrate, coordinating the interaction of successive ubiquitin linking (polyubiquitination) in order to induce their degradation. The function of the E3 ubiquitin-ligases may be reverted by deubiquitinating enzymes (DUB), by removing ubiquitin from substrate protein. Importantly, E1 and E2 enzymes have been minimally linked to cancer, while deregulation of the E3 ubiquitin-ligases has been extensively reported to play a role in cancer initiation and progression [43,44,45,46,47]. 

### 3.2. Classification of E3 Ubiquitin-Ligases

Due to their substrate specificity, E3 ubiquitin-ligases constitute a big and heterogeneous group of enzymes, with around six hundred types in humans. Therefore, each E3 ubiquitin-ligase is specific to a small subset of protein targets. They have historically been classified into two different families according to the specific domains and the mechanism used to transfer ubiquitin to the target protein: Homologous to E6-associated Protein C-terminus (HECT)-domain and Really Interesting New Gene (RING)-domain. The HECT-type enzymes receive ubiquitin from E2 enzymes and transfer it to the substrate. The RING-type domain E3 ubiquitin ligases, the most abundant family, can interact with the E2 enzyme to facilitate ubiquitin transferring to the protein substrate (Figure 2). [44].

HECT-type E3 can be subdivided into three main families: NEDD4, HERC and “other HECT” based on the domain architecture at N-terminal region [45,46,47,48]. The NEDD4 family is the best-characterized, consisting of nine human members: Smurf1, Smurf2, Itch, Wwp1, Wwp2, Nedd4, Nedd4-2, Hecw1, and Hecw2.

RING-type E3 ubiquitin-ligases have a varied architecture and they can be found as a single-chain enzyme, as homodimers and heterodimers (e.g., MDM2-MDMX), or as multi-subunit assemblies. A well-studied example of multi-subunit RING E3 ubiquitin-ligases is the Cullin RING ubiquitin ligase (CRL) superfamily. The CRL complexes combine a Cullin protein (Cul1, 2, 3, 4, 5 or 7), a RING finger protein (mainly Rbx1/Roc1 or Rbx1/Roc1), and an adaptor protein S-phase-kinase-associated protein-1 (Skp1) that binds to F-box proteins through their F-box domains. The assembled complex of the Skp1-Cullin1-F-box is named the SCF subfamily and contains 69 different F-box proteins [49,50]. Because of the structural complexity, the CRL superfamily recognizes an enormous range of proteins that can be targeted for ubiquitination, even without phosphorylated motifs [51]. Nevertheless, numerous RING-type proteins need phosphorylated-target proteins for recognition. Some of these E3 ubiquitin-ligases depend on phosphorylation at a tyrosine residue (pTyr), including three families: (i) SOCS-CL5-RING complex, with a typical SH2 recognition motif, (ii) Cbl family members and (iii) Hakai, containing a unusual pTyr-B domain, named HYB (Figure 3) [52,53].

In recent years, an increasing amount of studies have included other types of E3 ubiquitin- ligases. Some authors consider U-box (UFD2 homology) as a third subfamily. U-box ubiquitin-ligases act as scaffold proteins that recruit the ubiquitin-charged E2 enzyme and the substrate protein to docking sites [54,55]. Moreover, another novel family that shares features of RING and HECT is named RING-between-RING ligases (RBR). 

## 4. EMT Regulation by E3 Ubiquitin-Ligases 

As previously mentioned, E3 ubiquitin-ligases are frequently deregulated in human cancers, compared to E1 and E2 enzymes that have been minimally linked to cancer [43,44,45,46,47]. Indeed, abnormal expression, mutations or dysfunction of these enzymes have been associated with tumor progression. Given that the E3 ubiquitin-ligases confer substrate specificity, they are considered better therapeutic targets for cancer treatment than the E1, E2 enzymes or proteasome subunits themselves, since their inhibition would cause fewer side effects [56,57,58,59]. Still, every E3 ubiquitin-ligase targets an important number of specific substrates in different cells and contexts that could cause unforeseen consequences. Ongoing efforts are being made to develop anti-cancer drugs that targets E3 ubiquitin-ligases, testing the therapeutic effects in animal models as well as in clinical trials [43,60]. Targeted protein degradation is an emerging field that is expected to revolutionize how “undruggable” proteins can be targeted in diseases such as cancer. Indeed, it promises to be the greatest innovation in small-molecule drug discovery in the new millennium [61,62]. Cancer therapeutics targeting EMT-related E3 ubiquitin-ligases have been elucidated as a promising strategy to treat cancer, as they allow for the elimination of off-target side effects, compared to the strategy using anti-E1 and anti-E2, or an anti-proteasome subunit. Several E3 ubiquitin-ligases have been reported as being relevant during EMT by participating in the degradation of EMT-related proteins that directly or indirectly affect epithelial plasticity. These E3 ubiquitin-ligases, as well as the reported substrates and their functional roles, are summarized in Table 1 and are proposed as promising therapeutic targets. Moreover, a table of primary tumors where the E3 ubiquitin-ligases discussed are aberrantly expressed or mutated is also included (Table 2).

### 4.1. HECT-Domain E3 Ubqiuitin-Ligases

#### 4.1.1. Smurfs

Smad ubiquitination regulatory factor (Smurf) 1 and 2 are two members of the Nedd4 family of E3 ubiquitin-ligases, which contain a HECT catalytic domain. Various Nedd4 proteins are responsible for the ubiquitination of the epithelial sodium channel (ENaC) [134]. Nedd4, was shown to regulate IGF-IR ubiquitination and stability [135], suggesting its potential role in EMT. However, Smurf1 and 2 were identified based on their E3 ubiquitination activities on others targets. Smurf1 was originally reported to be an E3 ubiquitin-ligase for Smad1 and Smad5, which are intracellular signaling mediators for the Bone Morphogenetic Protein (BMP) pathway. On the other hand, Smurf2 interacts with Smad1 as well as Smad2, which are intracellular signaling molecules for TGF-β, inducing their ubiquitination and subsequent degradation [66]. Smurfs have several other different targets, some of them related with cell migration and metastatic potential, such as RhoA or Rap1B [50]. Due to the dual role of TGF-β/BMP signaling in cancer, Smurfs can act as tumor suppressors or promoters, depending on the cellular context. 

Smurf2 promotes the ability of tumor cells to metastasize by the regulation of a signaling pathway that is independent of Smads, Rap1B and RhoA in breast cancer cells [65]. Furthermore, Smurf2 expression was reported to be elevated in breast human cancer tissues and cell lines, and its stable overexpression promoted the ability of malignant MCF10CA1 breast cancer cells, derived from premalignant human breast epithelial MCF10A, to metastasize in an in vivo model. The ability of Smurf2-overexpressing cells to metastasize in vivo was correlated with EMT, since the overexpression of Smurf2 increases the level of N-cadherin even without TGF-β treatment. This suggests that Smurf2 may promote the EMT process by up-regulating N-cadherin expression [65]. On the contrary, Smurf2 can prevent cell migration by targeting other proteins, thus having the opposite effect in cancer progression. In this way, Smurf2 induces ubiquitin-dependent degradation of Smurf1 and prevents migration of breast cancer cells. Accordingly, knockdown of Smurf2 enhances Smurf1 levels, increasing breast cancer cell migration and metastasis in vivo [66]. TGF-β-induced EMT can also be suppressed by the sumoylation of Smurf2, which increases its ability to ubiquitinate TGF-β type I receptor (TβRI) in a three-dimensional culture of mammary epithelial cells [136]. Interestingly, Smurf2 is also regulated by TTC3, a RING-type E3 ubiquitin-ligases [67]. TTC3 is implicated in TGF-β1-induced EMT and myofibroblast differentiation, potentially through Smurf2 ubiquitination and degradation. 

Smurf1 is highly expressed in numerous tumor tissues and several experiments have demonstrated the metastasis-promoting role of Smurf1 in human cancer cells [102]. A recent report identified the role of Smurf1 in promoting cell migration and invasion capacities through the ubiquitination of DAB21P in ovarian cancer cells. DAB2IP, also known as ASK1-interacting protein-1 (AIP1), is a tumor suppressor that belongs to the Ras GTPase-activating protein (GAP) family. It is linked to cell-growth inhibition and apoptosis, and is down-regulated in various human cancers, promoting EMT and metastasis. The depletion of Smurf1 increases DAB21P expression and results in the impairment of the EMT process as well as cell proliferation and invasion in ovarian cancer cells [63]. Smurf1 also leads to the invasion and metastasis by the interaction and ubiquitination of Rho GTPase-activating protein 26 (ARHGAP26) in ovarian cancer cells [64]. Moreover, Smurf1 was reported to be negatively regulated by miR-1254, reducing EMT and PI3K/AKT signaling pathway in gastric cancer, further supporting its involvement in EMT [137]. These new reports propose Smurf1 as a potential therapeutic drug target in ovarian and gastric cancer, respectively. On the other hand, WWP2, a known HECT E3 ubiquitin-ligase, promotes Smad protein turnover and ubiquitination modulating TGFβ-dependent transcription and EMT. Selective disruption of WWP2 interactions with inhibitory Smad7 can stabilize Smad7 protein levels and prevent TGFβ-induced EMT [70]. Indeed, WWP1 associates to Smad7 inducing its nuclear export, and increases the binding of Smad7 to TGF-β type I receptor, causing ubiquitination and degradation of the receptor. Therefore, WWP1 negatively regulates TGF-β signaling in cooperation with Smad7. Interestingly, Smurfs and WWP1 are expressed in different patterns in human tissues, suggesting different roles in physiological and pathological conditions. Similar to Smurfs, Itch E3 ubiquitin-ligases positively regulate TGF-β signaling and EMT via Smad7 ubiquitination [68,69,138].

In sum, Smurf1 and Smurf2 are key proteins in the regulation of cancer metastasis, albeit further investigations are required to further elucidate their opposite roles in cell migration and invasion.

#### 4.1.2. Hect D1

Other HECT-type domain E3 ubiquitin-ligases have been reported to play a role in EMT. HectD1 (HECT domain E3 ubiquitin-ligase 1) was shown to polyubiquitinate ACF7 (also known as MACF1), inducing its degradation in a proteasome-dependent manner. ACF7 is a microtubule plus-end tracking protein (+TIP). TIPs are the only types of proteins that capture the reorganization of microtubule (MT) dynamics, which is essential to maintain EMT. Notably, +TIP ACF7 regulates cytoskeletal-focal adhesion dynamics and migration [139]. It was shown that the knockdown of HectD1 was directly related with an increased expression of ACF7 in breast cancer tumors, accompanied by a reduction in Vimentin expression and an increase in E-cadherin expression. Additionally, the depletion of HectD1 expression increased metastases formation in mouse models and conferred resistance to cisplatin. All these data suggest that ACF7 level is regulated by HectD1, thus modulating EMT and metastasis. Moreover, the increased expression of ACF7 as well as the reduced expression of HectD1 were reported to be correlated with poor clinical outcomes in breast cancer patients [72]. A recent study reported that HectD1 regulated Snail stability through ubiquitination, and this HectD1 depletion recovered Snail expression levels, thus impacting EMT. Additionally, they reported an association between lower levels of HECTD1 and a poor patient survival in human cervical cancer samples [71]. 

### 4.2. RING-Finger Domain E3 Ubqiuitin-Ligase

#### 4.2.1. SCF Family: F-Box Proteins

As previously mentioned, the CRL superfamily is the largest family of the multi-subunit RING-type E3 ubiquitin-ligases. Within this group, the SCF subfamily, constituted by Skp1-Cullin1-F-box proteins, has been very well studied. Over the last few years, some of the F-box proteins have been linked to EMT processes by their action on EMT-TFs [140]. Most EMT-TFs are short-lived proteins that are expressed at low levels in epithelial cells. Emerging evidence suggests that E3 ubiquitin-ligases may act both as tumor promoters or suppressors through the proteasome degradation of EMT-TFs and other specific substrates related to EMT. F-box proteins are the key recognition elements to determine which specific targets are ubiquitinated and degraded into the proteasome. E3 ubiquitin-ligases such β-TrCP1, Fbxl14, Fbxl5, Fbxo11 and Mdm2 are responsible for the ubiquitination and subsequent proteasome degradation of transcription factors such as Snail, Twist and Zeb [140], as recapitulated below.

The Fbxw7 has been categorized as a tumor suppressor due to promoting the degradation of several oncoproteins, such as Notch or c-Myc, which are potent inducers of the EMT [75,76]. Moreover, a study in cholangiocarcinoma cells demonstrated that Fbxw7 silencing promoted EMT, cell stemness and metastasis in vitro and in vivo [111] by the activation of mTOR and subsequent up-regulation of Snail, Slug and Zeb1. On the contrary, the mTOR inhibitor, rapamycin, blocked EMT, suggesting a potential link between Fbxw7 and Zeb1 through Fbxw7-mediated proteolysis of mTOR [111]. Furthermore, recent studies have proved that Fbxw7 can directly target Snai1 for degradation as well as ZEB2, which is degraded in GSK-3-mediated phosphorylation-dependent manner, thus suppressing EMT progress and chemoresistance [73,74].

Previous studies have demonstrated that Snai1 is also targeted for proteasome degradation by Fbxw1 (also named β-TrCP1). Fbxw1 is another member of the F-box subfamily of enzymes that mediates ubiquitination of Snail1, Snail2 in a GSK-3-mediated phosphorylation-dependent manner [78,141]. Interestingly, GSK-3β binds and phosphorylates Snail at two consensus motifs; phosphorylation of the first motif regulates its β-TrCP1-mediated ubiquitination, and phosphorylation of the second motif controls its subcellular localization. The inhibition of GSK-3β results in the up-regulation of Snail and down-regulation of E-cadherin in vivo. A recent important publication also demonstrated the action of A20 on the ubiquitination of Snail. A20 is an ubiquitin-editing enzyme [142] belonging to the deubiquitinating (DUB) type of enzymes. A20 plays an important role in inflammation and autoimmunity, but in 2017, Lee et al. demonstrated its role in TGF-β1-induced EMT and metastasis of basal-like breast cancers, through multi-monoubiquitylation of Snail1 [143]. A recent study showed that Fbxo16 also promotes the lysine 48-linked polyubiquitination of β-catenin, and its further degradation in the proteasome machinery, thus inhibiting EMT and preventing malignancy [89]. Fbxo11 also induces the ubiquitination and degradation of Snail1 in a Pkd1 phosphorylation-dependent manner, also impacting on EMT. Additionally, Fbxo11 promotes ubiquitin-mediated degradation of multiple Snail family members, including Snail1, Snail1 and or Scratch, a divergent subgroup of the Snail superfamily [144], during EMT and carcinoma progression as well as in mammalian epidermal development. Moreover, Fbxo11 expression was subject to down-regulation by TGF-β during the induction of EMT [87,88]. Fbxl5 and Fbxo31 were also reported to control Snai1 levels through its ubiquitination, resulting in the inhibition of cancer cell invasiveness. In addition, the role of Fbxl5 in cancer was reported in colon, breast and osteosarcoma [81,82,83,90].

However, the main regulator of the EMT process is Fbxl14, which regulates Snail1, Snail2, Twist1 and Zeb2. These EMT-TFs are functionally linked but not structurally related, suggesting that Fbxl14 may play a specific role in EMT process [84]. Fbx14 was demonstrated to induce Twist1 degradation in neck squamous carcinoma after treatment with imipramine blue (IB), a synthetic compound that inhibits EMT and stemness in HCHSCC cells, through the induction of Twist polyubiquitination and, subsequently, degradation in the proteasome [85]. The action of the IB compound on EMT through Fbx14 opens the possibility of identifying new compounds that may enhance the degradation of EMT-TFs mediated by F-box E3 ubiquitin-ligases. Fbxl4 induces the ubiquitination and degradation of CUB-domain-containing protein 1 (CDCP1), a transmembrane adaptor protein, identified as a malignant prognostic marker in several types of cancers, such as acute myeloid leukemia [145]. Indeed, Fbxl14 overexpression reversed EMT by decreasing CDCP1 levels, which in consequence inhibited EMT and metastasis in in vivo breast tumor mouse models [86]. Moreover, authors described miR-17/20a as a post-transcriptional regulator of Fbxl14, thus establishing Fbxl14 as an upstream regulator of the CDCP1 pathway.

Furthermore, Fbxo45 is an atypical E3 ubiquitin-ligase involved in EMT that is not able to form a complex with Cul1 due to the point mutation in its sequence. Instead, Fbxo45 binds to other proteins forming the complex Skp1–Pam–Fbxo45 [146]. Fbxo45 is implicated in the ubiquitination and further degradation of EMT-TFs including Snail1, Snail2, Twist1 and Zeb2 [91]. Negative regulation of Fbxo45 by miR-27a* resulted in EMT-TFs stabilization and therefore in the induction of the EMT process [91].

In summary, the E3 Ubiquitin-ligases family belonging to SCF play a crucial role in EMT by their action on a high number of substrates, including important EMT-TFs. For this reason, they have been proposed as potential anticancer drug targets.

#### 4.2.2. Mdm2

Murine double minute 2 (Mdm2) is a RING-type E3 ubiquitin-ligase that can form homodimers or heterodimers. Indeed, Mdm2 is one of the best-studied proteins in relation to cancer development, and is specifically up-regulated in cancers of mesenchymal origin. The best-studied substrate for Mdm2 is p53, whose recognition by Mdm2 leads to its polyubiquitination and proteasomal degradation. However, Mdm2 can inhibit p53 action by multiple mechanisms, being a primary cellular inhibitor of p53-mediated apoptosis [147]. Indeed, the inhibition of the interaction between p53-Mdm2 has been deeply investigated and it is considered an interesting strategy for cancer therapy. Mdm2 is considered a potent tumor promoter in numerous cancers and it has been highly associated with metastasis [148,149]. It was also described that Mdm2 ubiquitinates insulin-like growth factor type 1 receptor (IGF-1R), inducing its further degradation, thus affecting cell growth in a p53 independent-manner [150].

In recent years, a growing amount of evidence supports the involvement of Mdm2 overexpression in the induction of cancer EMT. It was proved that Mdm2 enhanced Snail expression and induced EMT in breast cancer cells, both in vitro and in vivo [151]. Snail expression induced by Mdm2 was accompanied by the regulation of other EMT markers, at mRNA and protein levels. It was also reported that Mdm2 facilitated cell motility and EMT in ovarian cancer cells though the positive regulation of TGF-β/Smads signaling, inducing the transcription of Snail1/Snail2 and the subsequent loss of E-cadherin levels [123]. Moreover, Mdm2 activates the Smad2/3 signaling pathway in lung cancer cells, thus promoting proliferation and the EMT process [122]. However, none of the above-mentioned publications showed a specific substrate for Mdm2 that was regulated by its E3 ubiquitin-ligase activity. Therefore, these findings indicate that the effect of Mdm2 on the suppression of EMT and tumor progression is not linked to its E3 ubiquitin-ligases activity. The first reported evidence pertaining to the regulation of EMT through the ubiquitin-ligase activity of Mdm2 was shown for Snail2 transcription factors. Authors demonstrated that Ling Zhi-8 (rLZ-8), recombinant protein of the medicinal mushroom *Ganoderma lucidum*, promoted the interaction between MDM2 and Snail2, resulting in Snail2 ubiquitination and degradation. rLZ-8 prevented lung cancer cell proliferation in an in vivo mouse model. The anti-metastatic activity of rLZ-8 by which it regulates EMT is through the negative modulation of FAK and Snail2 degradation. Therefore, rLZ-8 may be useful as a chemotherapeutic agent for treating lung cancer [92].

#### 4.2.3. TRIM Family

The tripartite motif (TRIM) proteins are defined as a subfamily of the RING-type E3 ubiquitin ligase family, harboring the RING finger and B-box as well as the coiled-coil (RBCC) domain motifs. Most of the TRIM proteins have E3 ubiquitin-ligase activity in their RING-finger domain. More than 80 TRIM proteins have been identified in humans and some of them have been associated with different cancers and EMT [152]. For instance, Trim 50 ubiquitinates and degrades Snail1. In a xenograft tumor model, TRIM50 showed an antitumor effect in hepatocarcinoma, as it acts as a tumor suppressor by directly targeting Snail1and reversing EMT. This suggests that Trim50 might be a novel therapeutic strategy for hepatocarcinoma treatment when Snail1 is overexpressed [93]. Moreover, Trim 62 (also named Dear1) was shown to block the TGF-β-SMAD3 signaling pathway, by the ubiquitination of Smad3, which results in the nuclear down-regulation of phosphorylated Smad3. Depletion of Dear1 increased Smad3 levels of downstream effectors Snail1 and Snail2 [94,153]. Many T Trim proteins have been shown to be up-regulated in different cancer tissues and to be related to the EMT process [154,155,156,157]. However, the exact mechanism by which they influence EMT is not well-known, although there is an increasing amount of evidence that suggests that it is not related to their ubiquitin-ligase activity. 

#### 4.2.4. Rbbp6 and Ppil2

Other less studied proteins have recently been described by their action on EMT through their E3 ubiquitin-ligase function. Rbbp6 is a RING finger-type E3 ubiquitin-ligase that induces the ubiquitination of IκBα protein, an inhibitor of NF-κB-signaling pathway. In consequence, it enhances the translocation of p65 to the nucleus and triggers the activation of NF-κB, thus inducing EMT and metastasis. NF-κB may activate EMT by up-regulation of N-cadherin, Vimentin and Snail1 in colorectal cancer cell lines [95]. Another E3 ubiquitin-ligase recently reported is Ppil2 (Peptidylprolyl isomerase (cyclophilin)-like 2). Ppil2 belongs to the U-box of E3 ubiquitin-ligases containing a modified RING-finger domain. It represses EMT breast cancer by inducing the ubiquitination of the Snai1 transcription factor. Moreover, it is suggested that Ppil2 might be involved in the immunosuppressant drug cyclosporine A (CsA)-mediated repression of EMT in breast cancer [96].

### 4.3. CBL Proteins 

Casitas B-lineage lymphoma (Cbl) proteins are a family of RING-type domain E3 ubiquitin-ligases that depend on phosphorylation of a tyrosine residue (pTyr) [52]. They contain several regions including: (1) an N-terminal tyrosine kinase binding domain (TKB domain, for p-Tyr residues of substrates); (2) a RING-finger domain (containing the intrinsic E3 ubiquitin-ligase activity); (3) a proline-rich region (involved in numerous SH3-domain interactions); (4) C-terminal ubiquitin-associated domain (UBA domain), that interacts with ubiquitin and ubiquitin-like domains of proteins [158,159].

As mentioned above, Cbl family members, including Cbl, Cbl-b and Cbl-c, recognize their substrates in a phosphotyrosine (pTyr)-dependent manner, and many receptor tyrosine-kinases (RTKs), including EGFR, MET, and RET, have been reported as Cbl-specific targets [160,161]. In 2012, a study using gastric and breast cancer cells and human samples showed that Cbl-b was a key factor in the epithelial phenotype maintenance, by the inhibition of cell migration in multi-drug resistant cancer cells in gastric and breast cancer tumors. Cbl-b is poorly expressed in this type of cancer cell and in gastric adenocarcinoma tissues, and its low expression correlated with increased tumor invasion and lymph node metastasis. The authors showed that Cbl-b overexpression induced ubiquitination and degradation of EGFR, thus preventing EMT and leading to the inhibition of the EGFR-ERK/Akt-miR-200c-ZEB1 axis [99]. In addition, Cbl-b also ubiquitinates and degrades IGF-IR. IGF-I-induced EMT in gastric cancer cells, accompanied by Zeb2 up-regulation, which involved the Akt/ERK-miR-200c-ZEB2 axis. Indeed, when Cbl-b induces degradation of IGF-IR, the inhibition of Akt/ERK-miR-200c-ZEB2 axis occurred, leading the repression of IGF-I-induced EMT [98]. As mentioned, other Cbl family members are involved in the regulation of EGFR. Indeed, it has been reported that the estrogen receptor beta-1 (ERβ1) induces the degradation of EGFR by enhancing the EGFR-Cbl interaction in basal-like breast cancer cells. ERβ1 inhibits EMT, invasion and migration by inducing the expression of E-cadherin through the up-regulation of the microRNA-200 family and repression of Zeb-1 and Sip-1 expression [97]. Moreover, a recent study showed that Cbl induces ubiquitination and degradation of EGFR by induced Pleckstrin-2 (Plek2). Plek2 expression is higher in gallbladder cancer compared to healthy adjacent tissues, where it promotes cell migration, invasion and liver metastasis in vivo, via the regulation of EMT. Mechanistically, Plek2 interacts with the kinase domain of EGFR and suppresses EGFR ubiquitination mediated by Cbl, leading to the constitutive activation of EGFR signaling [162]. All these data suggest the implication of Cbl ubiquitin-ligases in EMT by EGFR modulation in several cancer types.

Interestingly, Cbl contains mutations that are associated with human myeloid neoplasias in about 5% of the cases, and Cbl-c, exclusively expressed in epithelial cells, contains mutations associated to solid tumors. Overexpression of wild-type Cbl protein mediates the ubiquitination and down-regulation of the activated EGFR, while mutant Cbl-c fails to do so, acting as a dominant negative mutant. Indeed, the mutant Cbl-c binds to the EGFR and prevents the recruitment of the wild-type Cbl protein, suggesting that loss of Cbl-c is involved in cancer progression in solid tumors, both in mice and human models [161]. However, the connection of this mutant with EMT process awaits to be elucidated. 

#### Hakai

The E3 ubiquitin-ligase Hakai (also named Cbl-like-1 or Cbll1) was firstly identified as a RING-containing E-cadherin-binding protein that resembled the Cbl family [100]. However, Hakai is not a typical Cbl protein as it has different domains and mechanisms when it binds to pTyr residues in the target substrate [53]. Hakai possesses three different motifs: an N-terminal classic RING domain, a short pTyr recognition sequence, and a proline-rich domain in the C-terminal region, domains that are also harbored by the Cbl protein [100]. The RING-domain is directly adjacent to the HYB-domain, allowing the positioning of E2 and ubiquitin close to the pTyr residues in the substrate. Moreover, the mechanism of pTyr substrate binding is different between Hakai and Cbl, not being true homologues [25]. HYB-domain consists in a homodimer that forms a structurally different interface. Each monomer contains two zinc-finger domains between residues 106–148; an N-terminal RING domain followed by a short pTryrB regions that incorporate an atypical zinc-finger motif. Both motifs are important for dimerization [53]. For substrate recognition, the antiparallel dimerization of two monomers of Hakai is required. HYB-domain is responsible and critical for the interaction with targets, while the deletion of the RING-domain or the pTyr-domain deletion did not dramatically affect the association [25,53,163]. 

Hakai is the only reported E3 ubiquitin-ligase protein that is involved in the direct regulation of E-cadherin at cell–cell contacts, a hallmark of the EMT [53,100]. Hakai binds to the cytoplasmic region of Src tyrosine-phosphorylated E-cadherin and promotes its ubiquitination, endocytosis and degradation [100,164]. Although additional interacting proteins were reported for Hakai, to date only its action on E-Cadherin has been directly related to EMT [165,166]. Hakai is not only involved in the E-cadherin down-regulation at cell–cell contacts, it is also implicated in cell proliferation, invasion and metastasis in vitro and in vivo [25,101,167,168,169,170].

Moreover, Hakai is highly expressed in lung and colon cancer tissues compared to adjacent normal tissues [38,101,133]. Given that the loss of E-cadherin is a probably the best characterized hallmark of EMT, Hakai is considered as a promising therapeutic target against cancer [24]. Indeed, the first specific small-molecule inhibitor for Hakai, named Hakin-1 (Hakai-inhibitor 1), has been reported recently, which was specifically designed to target the HYB domain of Hakai, where phosphorylated-E-cadherin specifically binds. Hakin-1 inhibits Hakai-dependent ubiquitination of E-cadherin reverting the EMT process. Hakin-1 was shown to inhibit carcinoma growth and tumor progression in vitro, in colorectal cancer cell lines, and in vivo, in a tumor xenograft mouse model, without apparent systemic toxicity in mice. These results represent an important step forward in the small-molecule drug discovery that may impact the future development of drugs targeting E3 ubiquitin-ligases as a strategy for anti-metastatic cancer therapies.

## 5. Conclusions

Around 90% of cancer-deaths are due to metastasis, a multistep process that enables primary tumors to disseminate, invade and colonize distant organs. The epithelial-to-mesenchymal transition (EMT) is a central player in carcinoma metastasis, representing around 80% of cancers. During EMT, tumor cells increase migratory and invasive potential. Perhaps the best characterized example of EMT is the loss of the E-cadherin, a cell–cell adhesion molecule, accompanied by the expression of mesenchymal markers such as N-cadherin or Vimentin. EMT is described to be regulated by transcription factors such as Snail1, Snail2, Twist or Zeb. Moreover, EMT is triggered in response to different signals including TGF-β and EGF. In recent years, protein degradation by polyubiquitination process has been considered as a promising target against cancer. Indeed, targeting E3 ubiquitin-ligases is anticipated to be a better option as these confer substrate specificity, therefore less toxicity is expected. In this review, we have presented the reported E3 ubiquitin-ligases that directly or indirectly control EMT in cancer. 

Several E3 ubiquitin-ligases are reported to indirectly affect EMT through the action on target stimuli and signaling pathways associated with EMT, such as Surf1, Smurf2, Ttc2 WWP2, Trim62 or Cbl and Cbl-b. However, not many direct regulators of the reported hallmarks of EMT have been described. In fact, the most reported E3 ubiquitin-ligases influencing EMT are those affecting EMT-TFs. As the EMT-TFs are very unstable proteins, they have to be stabilized for EMT to progress. Many F-box proteins are reported to target EMT-TFs. F-box E3 ubiquitin-ligases such as Fbxw1, Fbxw7, Fbxl14 and Fbxo11 are reported to target Snail1 and other EMT-TFs, for ubiquitination and subsequent degradation, therefore inhibiting EMT and preventing metastasis. Another example is the HECT-finger domain E3 ubiquitin-ligase, HectD1 that also targets Snail1 EMT-TF. Apart from this direct effect on EMT-TFs, the only reported E3 ubiquitin-ligase involved in the direct regulation of E-cadherin at cell–cell contacts is Hakai. Hakai is a novel Cbl subfamily that binds tyrosine-phosphorylated E-cadherin in a Src dependent manner. Hakai promotes E-cadherin ubiquitination, endocytosis and degradation, directly impacting on EMT in vitro and in vivo. Considering that the loss of E-cadherin is a hallmark of EMT, Hakai is considered as a promising therapeutic target against cancer. Moreover, Hakai contains a novel HYB domain that is structurally different from other phosphotyrosine-binding domains and is proposed as potential drug target due to its novel structural features. Recently, Hakin-1 was reported as the first small-molecule inhibitor for Hakai specifically designed to target the HYB domain, where phosphorylated-E-cadherin specifically binds. Hakin-1 inhibits Hakai-dependent ubiquitination of E-cadherin reverting EMT process and inhibiting carcinoma growth and tumor progression in vitro and in vivo. Further studies are required to advance into the unearthing of novel small-molecule drugs targeting EMT-related E3 ubiquitin-ligases. Small-molecule drug discovery in this field will impact future investigations, and testing their therapeutic effects in animal models and in clinical trials will impact on anti-metastatic cancer therapies. 

## Figures and Tables

**Figure 1 cancers-12-03093-f001:**
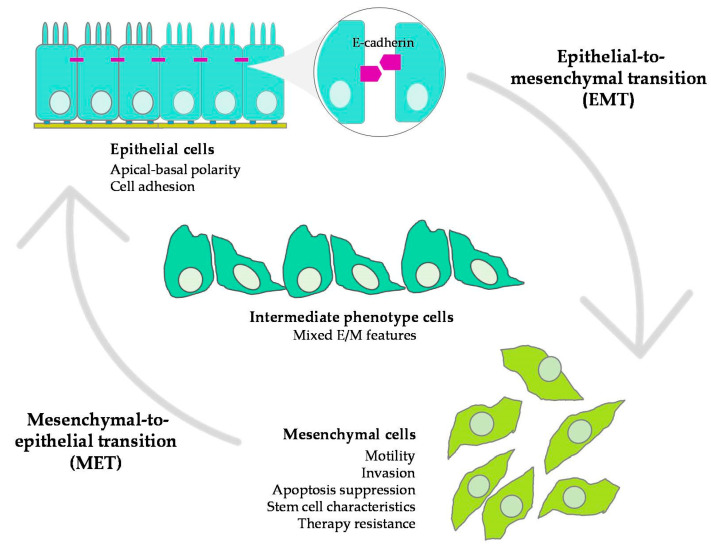
An overview of epithelial–mesenchymal plasticity (EMP). During epithelial-to-mesenchymal transition (EMT), epithelial cells undergo phenotypic changes into mesenchymal phenotype, accompanied by the loss of cell–cell contacts and apical-basal polarity. Intermediate phenotypic states, between epithelial and completely mesenchymal states may coexist, resulting in mixed E/M features. EMT is a reversible process and the resulting mesenchymal cells can revert into an epithelial phenotype by mesenchymal-to-epithelial transition (MET).

**Figure 2 cancers-12-03093-f002:**
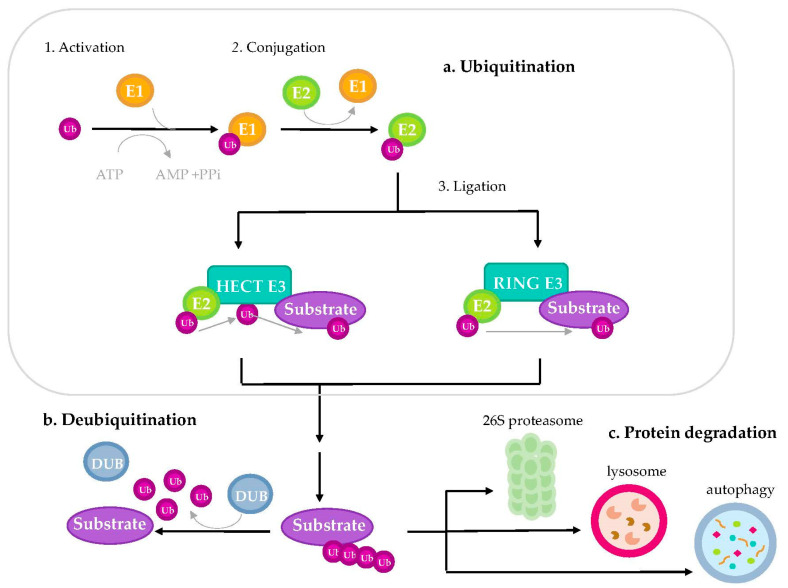
Ubiquitination cascade involved in targeted protein degradation. (**a**) Ubiquitination process. 1. Activation: ubiquitin (Ub) molecule is activated by the ubiquitin-activating enzyme (E1) in an ATP-dependent manner. 2. Conjugation: the ubiquitin-conjugating enzyme (E2) transfers the activated ubiquitin to the ubiquitin-ligase enzyme. 3. Ligation: the ubiquitin-ligase (E3) responsible for transferring the ubiquitin to the target substrate. E3 ligases can be classified in two major types, Homologous to E6-associated Protein C-terminus (HECT) E3 or Really Interesting New Gene (RING) E3, which differ in the mechanism used to transfer the ubiquitin (**b**) Deubiquitination: ubiquitin moieties can be removed from substrate proteins by deubiquitinase enzymes (DUB); (**c**) Protein degradation: ubiquitinated proteins are degraded by the proteasome, lysosome or autophagy.

**Figure 3 cancers-12-03093-f003:**
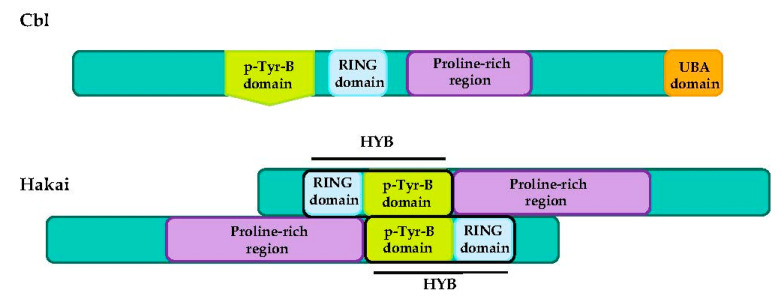
Schematic diagrams of typical RING-finger type E3 ubiquitin ligases (Cbl, upper panel) compared to the E3 ubiquitin-ligase Hakai protein structures (Cbll1) that forms a homodimeric structure, described by Mukherjee et al., 2012 [53].

**Table 1 cancers-12-03093-t001:** E3 ubiquitin-ligases involved in EMT as promising drug targets in cancer. The reported substrates and their functional roles are also summarized.

Protein	EMT-Related Substrates	Functional Roles	Reference
Smurf 1	Smad1, Smad2, DAB21P, ARHGAP26	Promotes cell migration and invasion and metastasis	[63,64]
Smurf 2	Smad1, Smad2, Smurf1, TβRI,	Up-regulation of N-cadherin. Knockdown of Smurf2 enhances Smurf1 levels, increasing migration and metastasis in vivo	[65,66,67]
Ttc3	Smurf2	Implicated in TGF-β1-induced EMT	[67]
Itch	Smad7	Positively regulates EMT induced by TGF-β	[68]
WWP1	TGF- β type I Receptor	Negatively regulates TGF-β tumor suppressor pathway	[69]
WWP2	Smads	TGFβ-dependent transcription and EMT	[70]
HectD1	ACF7, Snail	Inhibits the reorganization of microtubule and promote epithelial phenotypes	[71,72]
Fbxw7	Zeb1, Snai1, c-myc, Notch	Tumor suppressor. Suppression of EMT and chemoresistance	[73,74,75,76]
Fbxw1 (β-TrCP1)	Snai1, Snail2	Oncogenic or tumor-suppressor functions in a tissue-specific manner	[77,78,79,80]
Fbxl5	Snai1	Inhibition of gastric cancer metastasis	[81,82,83]
Fbxl14	Snai1, Snail2, Zeb2, Twist, CDCP1	Suppression of EMT, stemness and metastasis	[84,85,86]
Fbxo11	Snai1, Snail2, Scartch	Blocks EMT in breast cancer. Down-regulated by TGF-β	[87,88]
Fbxo16	β-catenin	Inhibits EMT and prevent malignancy	[89]
Fbxo31	Snai1	Inhibition of gastric cancer metastasis	[90]
Fbxo45	Snai1, Snail2, Zeb2, Twist	Suppresses EMT characteristics. Negative regulated by miR-27a*	[91]
Mdm2	Snail2	rLZ-8,a recombinant protein of medicinal mushroom *G. lucidum*, promoted the degradation of Snail1 by MDM, preventing lung cancer cell proliferation in vivo	[92]
Trim50	Snail1	Antitumor and anti-EMT effect in hepatocarcinoma	[93]
Trim62	SMAD3	Depletion of EMT TFs	[94]
Rbbp6	IκBα	Activation of NF-κB-and EMT	[95]
Ppil2	Snai1	Repression of EMT in breast cancer	[96]
Cbl	EGFR	Inhibition of EMT, invasion and migration by the degradation of EGFR in response to ERβ1	[97]
Cbl-b	EGFR, IGF-IR	Maintenance of the epithelial phenotype by inhibition of cell migration in gastric and breast multidrug resistant cancer cells	[98,99]
Hakai	E-cadherin	Disruption of cell–cell contacts and induction of EMT in vitro and in vivo	[58,100,101]

**Table 2 cancers-12-03093-t002:** Primary tumors where the discussed E3 ubiquitin-ligases involved in EMT are mutated or aberrantly expressed.

Protein	Associated Cancers	Reference
Smurf 1	Overexpressed in colorectal and gastric cancer	[102,103]
Smurf 2	Overexpressed in esophageal squamous cell carcinoma	[104]
Itch	Overexpressed in breast cancer, lung cancer and amplification in 20q11.22 in thyroid carcinoma	[105,106,107]
WWP1	Overexpressed in prostate cancer and breast cancer	[108,109]
HectD1	Downregulated in breast cancer and cervical cancer	[71,72]
Fbxw7	Inactivated by mutation in bile duct, blood, endometrium, colon and stomach cancer.	[110]
	Downregulated in cholangiocarcinoma	[111]
Fbxw1 (β-TrCP1)	Expression increased in colorectal cancer	[112]
Fbxl5	Downregulated in gastric cancer	[83]
Fbxl14	Downregulated in breast cancer	[86]
Fbxo11	Deleted or mutated in diffuse large B-cell lymphoma	[113]
	Expression decreased in skin cancer, glioblastoma and prostate cancer	[114]
	Reduced expression is correlated with adverse clinical outcome in lung cancer	[88]
Fbxo16	Expression attenuated with cancer progression in breast cancer	[89]
	Downregulated in glioblastoma	[115]
Fbxo31	Downregulated in gastric cancer, breast cancer and hepatocellular	[116,117,118]
Fbxo45	Overexpressed in squamous-cell lung carcinoma (SCLC)	[119]
	Downregulated in gastric cancer	[120]
Mdm2	Overexpressed in leukemias and on-third of sarcomas, lung cancer and ovarian cancer	[121,122,123]
Trim50	Downregulated in hepatocarcinoma and ovarian cancer	[93,124]
Trim62	Mutated and downregulated in breast cancer	[125]
	Downregulated in non-small cell lung cancer (NSCLC)	[126]
Rbbp6	Overexpressed in esophageal cancer, breast cancer, lung cancer, cervical cancer and colorectal cancer	[95,127,128,129,130]
Ppil2	Downregulated in breast cancer	[96]
Cbl	Downregulated in lung cancer	[131]
Cbl-b	Downregulated in gastric and breast cancer	[98,99]
	Associated with better outcome in NSCLC patients	[132]
Hakai	Overexpressed in colorectal cancer and NSCLC	[101,133]

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
