# Peer review of "Regulation of Epithelial–Mesenchymal Plasticity by the E3 Ubiquitin-Ligases in Cancer"

_cancers, 2020, doi:10.3390/cancers12113093_

Round 1

Reviewer 1 Report

Rodríguez-Alonso and colleagues present a review article of epithelial-mesenchymal plasticity associated with E3 ubiquitin ligases in malignancy.

I would like to see some more reasoning behind statements of why certain E3s might be good targets to inhibit. It would be helpful to include a table of primary tumours where the E3s discussed are aberrantly expressed or mutated. There could be less emphasis on some of the standard more textbook interactions such as the ubiquitin cascade, and more emphasis on the function of specific E3s that could be shown with schematics.

English needs to be addressed throughout. I have made some corrections below, but have not flagged all instances.

Minor Ammendments

  1. There is a little confusion in the below and perhaps a little repetition regarding what this review is going to be focusing on exactly?

ln 123-127, “In the last decades, the control of the EMT at transcriptional, post-transcriptional and posttranslational levels (by phosphorylation, glycosylation or proteolysis) [39,40] has been very well documented [18,29], therefore in this review we will go in depth. However, in the recent years, post translational regulation of EMT by ubiquitination process has become increasingly relevant, therefore in the present review, we will go deeper into it”

  1. This statement negates to mention the roles of ubiquitination in the control of transcription (e.g. histones H2B and H2A including the Polycomb Repressor Complex) and the facilitating of DNA damage repair. This also then impacts on Figure 2 in that its context needs to be broadened. The authors should consider differentiating “polyubiquitination” and “monoubiquitination” given that end results are different in the context of degrading / getting rid of proteins (and this is really what this statement and figure are about).

ln 130 – 132, “Ubiquitination is a reversible post-translational modification consisting on the conjugation of a 76-amino acid ubiquitin molecule to target protein for its degradation into proteasome, lysosome or autophagy (Figure 2)”

  1. Same comment as above. A statement like this is negating other roles of ubiquitination and especially those associated with monoubiquitination.

ln 133, “Although ubiquitination process generally targets cytosolic or nuclear proteins for degradation  via proteasome, many membrane proteins undergo degradation into lysosomes.”

  1. ln 143, “enzyme and subsequently transfer it to a specific signalized substrate …” I am not sure of the phrase “signalised substrate”?

  1. ln 144, “The function of the E3 ubiquitin-ligases may be reverted by deubiquitinases deubiquitinating enzymes (DUBs), by removing ubiquitin from the substrate protein …”

  1. ln 150 – 151 “Due to their substrate specificity, E3 ubiquitin-ligases constitute a big and heterogeneous group of enzymes, existing about of which there are around six hundred in humans …”

  1. ln 156 – 157, Not sure what is meant here by “set close” … “The RING-type domain, the most abundant family, set close to E2 enzyme to facilitate ubiquitin transferring to the protein substrate (Figure 3) [48]. …”

  1. ln 162, The comment is made earlier that there are around 600 E3 ligases, but here the comment is made pertaining only to the RING E3s (ie. not the HECT). Which is correct? – “The mammalian genome encodes more than 600 potential RING finger E3s [48,53]”

  1. ln 181, This sentence does not make sense: “Despite the U-box and RBR ligases are considered as different subfamilies of E3 ubiquitin-ligases [55,60], both of them have been also categorized into  RING-type ligases family [54,61,62].”

  1. Ln 187 – 189, “Moreover, given that the E3 ubiquitin-ligases confer substrate specificity, they are considered promising therapeutic targets for cancer treatment, since their inhibition would cause less side effects than the inhibition of E1, E2 enzymes or proteasome subunits itself themselves”(?)

  1. The MCF10Ca1a cell line should be explained as per ref #70. While most in the field would be familiar with MCF-10A cells, this metastatic line would be less familiar (ln 218).

  1. This sentence does not make sense (ln 244) “On the other hand, WWP2, a HECT E3 ubiquitin-ligase known is novel Smad binding partner, thus modulating TGFβ-dependent transcription and EMT.”

  1. ln 252-54 When it is polyubiquitination, this should be clearly stated, as not all “ubiquitination” leads to proteasomal degradation.     “Other HECT-type domain E3 ubiquitin-ligases have been reported to play a role in EMT. HectD1 (HECT domain E3 ubiquitin-ligase 1) was shown to ubiquitinates ACF7 (also known as MACF1) inducing its degradation in a proteasome-dependent manner.”

  1. ln 253 “…was shown to ubiquitinates ACF7 … “ Again if polyubiquitinates then state this.

  1. ln 272 “… In Over the last few years, some of the F-box proteins have been related with to EMT processes by its their action on EMT-TFs [111]. … “

  1. ln 284, “ … and metastasis in vitro and in vivo [112], by the activation of mTOR and the subsequently the up … “

  1. ln 311, “… However, the main regulator of the EMT process is Fbxl14, which regulate Snail1, Snail2, Twist1 …”

  1. ln 319, Please give examples. “CDCP1 (CUB-domain-containing protein 1), a transmembrane adaptor protein, that was identified as a malignant prognostic marker of several tumors …”

  1. ln 338, “Mdm2 leads to its polyubiquitination and proteasomeal degradation … “

  1. ln 343, “ … associated to with metastasis [119,120]….”

  1. ln 354, what does this mean “Mdm2 is independent of its E3 ligase-ligases activity”

  1. ln 365, “More than 80 TRIM proteins have been identified in humans and some of them have been associated to different cancers and 367 EMT [124]”

  1. ln 384 – 387, Are “Ppil2” and “Ppl2” used interchangeably?

  1. ln 452, “Given that the loss of E-cadherin is a probably the best characterized 453 hallmark of EMT …”

  1. ln 470, Need to be indicating that this is polyubiquitination – “In recent years, protein degradation by ubiquitination …”

  1. ln 496, I don’t think that you want to be disrupting future investigations. Please rephrase. “Small-molecule drug discovery in this field will disrupt future investigations, and testing the therapeutic effects in animal models and in clinical trials will impact on anti-metastatic cancer therapies.”

Typographical errors

ln 91, “Given the well-r role of EMT in tumor”

ln 138-140, “The E3 ubiquitin-ligase enzyme is responsible for specific binding of ubiquitin to the target substrate [41,42]; the mechanism used to transfer ubiquitin to the target depends on the specific domain they contain.”

ln 140, In that sense, substrate specificity of the ubiquitin  proteasome system relies on the E3 ubiquitin ligases, which are larger in number in comparison with …”

ln 152, “They have historically been classified into two different families … “

ln 165, “example of multi-subunit RING E3 ubiquitin-ligases is the Cullin RING ubiquitin ligase (CRL) …”

ln 186-187, “Indeed, abnormal expression, mutations or dysfunction of these enzymes have been associated to with tumor progression”

ln 189 – 191, “Ongoing efforts are made to develop anti-cancer drugs that targets E3 ubiquitin-ligases, testing the therapeutic effects in animal models as well as in clinical trials [42,66]”

ln 205-206 “Smad ubiquitination regulatory factor (Smurf) 1 and 2 are two members of Nedd4 family E3 206 ubiquitin-ligases, which contain a HECT catalytic domain”

ln 206, “Various Nedd4 proteins are responsible of for the ubiquitination of epithelial sodium channel (ENaC) [105].”

ln 212, “Smurfs have several other different targets, some of them related to with cell … “

ln 215, “Smurfs2 promotes the ability of tumor cells to metastasize …”

ln 232, “A recent report identified has reported the role of Smurf1 in promoting cell migration and invasion capacities…”

ln 236, “It is related with to cell-growth inhibition and apoptosis, and is down-regulate …”

ln 238, “…expression and resulted in the impairing impairment of EMT process…”

ln 243, “…These new reports propose Smurf1 as a potential therapeutic drug target…”

ln 265, “ … and this HectD1depletion recovered … “ space missing

ln 276, “ … F-box proteins are the key recognition elements to determinate determine which …”

ln 312, “These EMT-TFs are functionally linked but not structurally related … “

ln 315, “inhibits E;T” …. should this be EMT?

ln 316, “ubiquitination and subsequently in the proteasome … “

ln 344, “In the last years, a growing amount of evidences supports that Mdm2 overexpression is involved …”

ln 347, “I [It?] was also 348 reported that Mdm2 facilitated cell motility and EMT in ovarian cancer cells though the positive 349 regulation of TGF-β/Smads signaling, inducing the transcription of Snail1/Snail2 and the subsequent 350 loss of E-cadherin levels [122] …”

ln 488, “loss of E-cadherin is a hallmark of EMT … “

Author Response

Specific comments to Reviewer #1:

We thank the reviewer for the insightful comments. Here, we provide an answer to each of the concerns.

I would like to see some more reasoning behind statements of why certain E3s might be good targets to inhibit. It would be helpful to include a table of primary tumours where the E3s discussed are aberrantly expressed or mutated. There could be less emphasis on some of the standard more textbook interactions such as the ubiquitin cascade, and more emphasis on the function of specific E3s that could be shown with schematics.

- Firstly, as the reviewer suggested we have added more reasoning behind statements of why certain E3s might be good targets to inhibit (page 6, lines 188-190; 199-202). Moreover, in order to put more emphasis on the function of specific E3 ubiquitin-ligases on cancer-EMT, we have removed figure 3 as schematics, and we have included a table of primary tumors where the E3 ubiquitin-ligases discussed are aberrantly expressed or mutated (page 7-8, lines 210-211 and Table, 2).

English needs to be addressed throughout. I have made some corrections below, but have not flagged all instances.

- As the reviewer recommended, we have properly revised the grammar and spelling errors to get an easy reading. For this purpose, an English native has checked our manuscript´s language to the requiered level by international journals.

Minor Amendments

  1. To avoid confusions, we have revised and corrected the sentence in page 4, lines 121-124.
  2. Thank you for the careful revision. As the reviewer suggested, we have broadened the context of the ubiquitination process with a general role, Moreover, as suggested, we have included the differentiation between “monoubiquitination, multi-monobiquitination and polyubiquitination” (page 4 lines 126-136).. Furthermore, we have corrected the figure legend 2 to only focus on targeted protein degradation mediated by ubiquitination (Figure 2, figure legend 2)
  3. Thank you very much for the important point suggested. We have broadened the context of the ubiquitination process with a general role (page 4 lines 126-136), and to avoid misunderstanding we have removed the indicated sentence.
  4. As the reviewer suggested we have removed the term “signalized substrate”? (line 144)
  5. We have corrected the word using “deubiquitinating enzyme” (line 146)
  6. We have corrected the sentence using “which there are around”(line 152)
  7. The sentence in lines 157-158 was revised.
  8. We have removed the sentence “The mammalian genome encodes more than 600 potential RING finger E3s [48,53]”. The correct sentence is in lines 151-152. Thank you for the careful correction
  9. We have removed the confusing sentence “Despite the U-box and RBR ligases are considered as different subfamilies of E3 ubiquitin-ligases, both of them have been also categorized into RING-type ligases family”.
  10. Line 192, we have added the word “themselves.
  11. Clarification of the type of cell line is MCF10Ca1a is included in line 230-231.
  12. The sentence in line 256-258 was corrected. Thank you for the careful revision.
  13. It was clearly specified that there is “polyubiquitination” (line 269-270)
  14. In line 270, it was clearly specified the “polyubiquitination” of ACF7.
  15. The sentence was properly corrected (line 289-290).
  16. Sentence in line 300 was corrected.
  17. It was added “the” before EMT in line 329.
  18. It was included an example on which CDCP1 is prognostic marker (lines 337-338).
  19. The sentence “Mdm2 leads to its polyubiquitination and proteasomeal degradation” was corrected (line 256)
  20. In line 360, the sentence “associated with metastasis” was corrected.
  21. The sentence in lines 373-374 was corrected.
  22. In 386, the word “identified” was included.
  23. The correct name is Ppil2 (corrected in lines 397, 403 and 406). Thank you for the profound correction.
  24. Sentence in line 472 was corrected.
  25. ln 491, the word “polyubiquitination” is indicated,
  26. We have use “impact” word instead of “disrupt” (line 516). Thank for the important corrections all over the text

All typographical errors were revised and corrected.

ln 89-90, “well-recognized”

ln 140, Included “for” in the sentence

ln 142, Included “the” in the sentenceIn that sense, substrate specificity of the ubiquitin  proteasome…”

ln 153, this sentence “They have historically been classified into two different families … “ was corrected

ln 170, this sentence “example of multi-subunit RING E3 ubiquitin-ligases is the Cullin RING ubiquitin ligase (CRL) …” was corrected.

ln 191, this sentence “Indeed, abnormal expression, mutations or dysfunction of these enzymes have been associated to with tumor progression” was corrected.

ln 196, this sentence “Ongoing efforts are made to develop anti-cancer drugs that targets E3 ubiquitin-ligases, testing the therapeutic effects in animal models as well as in clinical trials [42,66]”was revised.

ln 216 this sentence “Smad ubiquitination regulatory factor (Smurf) 1 and 2 are two members of Nedd4 family E3 206 ubiquitin-ligases, which contain a HECT catalytic domain” was corrected.

ln 217, this sentence “Various Nedd4 proteins are responsible of for the ubiquitination of epithelial sodium channel (ENaC) [105].” was corrected.

ln 224, this sentence “Smurfs have several other different targets, some of them related to with cell … “was corrected.

ln 227, “Smurfs2” was changed to “Smurf2”

ln 246, This sentence “A recent report identified has reported the role of Smurf1 in promoting cell migration and invasion capacities…” was corrected.

ln 249, This sentence “It is related with to cell-growth inhibition and apoptosis, and is down-regulate …” was corrected.

ln 251, This sentence “…expression and resulted in the impairing impairment of EMT process…” was corrected.

ln 255, This sentence “…These new reports propose Smurf1 as a potential therapeutic drug target…” was corrected.

ln 282, “ … and this HectD1depletion recovered … “ space was included

ln 294, This sentence “ … F-box proteins are the key recognition elements to determinate determine which …” was corrected.

ln 330, This sentence “These EMT-TFs are functionally linked but not structurally related … “was corrected.

ln 333, This sentence “inhibits E;T” …. was corrected.

ln 323, This sentence “ubiquitination and subsequently in the proteasome … “was corrected.

ln 364-365, This sentence “In the last years, a growing amount of evidences supports that Mdm2 overexpression is involved …” was corrected.

ln 367 This sentence “I [It?] was also 348 reported that Mdm2 facilitated cell motility and EMT in ovarian cancer cells though the positive 349 regulation of TGF-β/Smads signaling, inducing the transcription of Snail1/Snail2 and the subsequent 350 loss of E-cadherin levels [122] …” was corrected.

ln 508, This sentence “loss of E-cadherin is a hallmark of EMT … “was corrected.

Reviewer 2 Report

My main comment for the manuscript is that it needs a fair amount of modification with regards to English language and grammar.  In particular, there are numerous instances of subject-verb disagreement, improper use of articles like "a/an" and "the," and awkward phrasing that detracts from a very important summary in the field.  An example of the latter is the last paragraph in section "3. Molecular Mechanisms of EMT in cancer: ubiquitination."  This paragraph is meant to justify the focus of this review as looking exclusively at the post-translational regulation of EMT due to previous work looking at transcriptional and post-transcriptional regulation, but the way it is currently written suggests otherwise.

Some other points that the authors should consider on revision:

1) Define E/M on first usage

2) Although the off-target effects of a cancer therapy targeting an E3 ligase would be less than that of an anti-E1, and anti-E2, or an anti-ribosomal subunit approach, it must still be acknowledged that these E3's target 10's if not 100's of proteins (many of which may not be part of the EMP state) in different cells and in different contexts.  Therefore, even this approach could have unforeseen consequences.

3) In terms of HECT E3's, both ITCH and WWP1 have also been shown to regulate TGF-beta signaling and influence cancer initiation and promotion in vivo. How would they tie in to the picture put forth here? 

4) How does redundancy of targets among E3 ligases (not just similar family members but also between HECT and RING family members) confound the potential for using them as anti-cancer targets?

Author Response

Specific comments to Reviewer #2:

We sincerely appreciate the referee´s careful review of our manuscript.

Overall, as the Reviewer #2 recommended, we have carefully revised the language and grammar and spelling errors to improve the manuscript. For this purpose, a native English speaker has checked our text and edited our manuscript´s language to the requiered level by international journals. Moreover, in order to avoid confusion by the readers we have justified the foccus of this review in targeted protein degradation by ubiquitination (page 3, line 92 and page 4, line 121-124).

The rest of the minor points indicated by reviewer #1 are clarified as follow:

1) We have defined E/M on first usage (Page 2, line 67)

2) We fully agree with the reviewer regarding to the comment on the off-target effects when targeting E3 ubiquitin-ligases. As suggested, we have revised the statement (page 6, lines 194-196).

3) Thank you for the careful revision. As the reviewer indicated we have discussed the role of ITCH and WWP1 in regulation of TGF-beta signalling (page 9, lines 259-265). Moreover, we have also included them in Table 2.

4) As discussed in the manuscript, we propose the that the use of E3 ubiquitin-ligases as anticancer drug targets may be better approach than the use of the E1, E1 or proteasome system, given that the specificity of the ubiquitination system relies in them. However, some of the E3 ubiquitin-ligases share common targets, therefore the inhibition on a specific substrate can be compensated by the activity of other ligases on the same substrate. Thus, redundancy of targets among E3 ubiquitin-ligases can be a limitation as using a specific inhibitor may not cause the decrease of a target degradation, as other E3 ubiquitin-ligase may induce it. Therefore, more investigations regarding to possible compensation are required.

Reviewer 3 Report

The epithelial-mesenchymal plasticity (EMP) is a process on which epithelial cells acquire the ability to dynamically switch between epithelial and mesenchymal phenotypic cellular states. This plasticity has been very well described during epithelial-to-mesenchymal transition (EMT) in cancer. During EMT, the conversion from an epithelial phenotype to a mesenchymal phenotype confers increased cell motility, invasiveness and the ability to disseminate to distant sites and to form metastasis. Given the crucial role of EMT during tumor progression, the modulation of molecularly defined targets involved in this process has become an attractive therapeutic strategy against cancer. Targeted protein degradation promises to be the greatest innovation in small-molecule drug discovery in the new millennium. Protein degradation carried out by ubiquitination has gained attention to selectively degrade proteins of interest. In this review, we summarize 29 the role of the E3 ubiquitin-ligases that control EMT during cancer progression, and we highlight 30 the potential use of the E3 ubiquitin-ligases as drug targets for the development of small-molecule 31 drugs against cancer.

The review is interesting and well-written and covers the literature well. I just have some minor suggestions to improve the manuscript.

  • The Abstract is a little long. I would cut it a bit to focalize on a couple of key issues.
  • One aspect that can be better pointed out is that EMT is definitely connected to the initial stage of the metastatic process. Cancer cell motility from the primary site is definitely associated with EMT as is the invasive process. However, when cancer cells reach the secondary site usually revert to an epithelial phenotype to enhance cell adhesion to the substrate of the secondary site and form metastasis (MET process). This plasticity can be better highlighted in the text,
  • The role of Cbl-b in regulating IGF-IR-dependent EMT has been well described. However, Additional E3 ligases have been shown to target the IGF-IR, including Nedd4 and MDM2. It is therefore important to brief mention the fact that these ligases (Nedd4 and MDM2) may be indirectly affect EMT by regulating IGF-IR levels.

Author Response

Specific comments to Reviewer #3:

We appreciate the constructive reviewer´s comments. As the reviewer

  • As the reviewer suggested we have shorten up the abstract and focus in the most important issues (Page 1, lines 19-29).
  • As the reviewer suggested we have better highlighted the “MET” process in the text (page 2, lines 58-61).
  • As indicated by the reviewer, we have briefly mentioned the role of Nedd4 and MDM2 on EMT by regulating IGF-IR levels (page 8, lines 217-218 and page 11, lines 361-263)